# Discrete spatial organization of TGFβ receptors couples receptor multimerization and signaling to cellular tension

**Joanna P Rys[1,2†], Christopher C DuFort[2†], David A Monteiro[1,2], Michelle A Baird[3], Juan A Oses-Prieto[4], Shreya Chand[4], Alma L Burlingame[4], Michael W Davidson[3], Tamara N Alliston[1,2,5]\***

[1]UC Berkeley-UC San Francisco Graduate Program in Bioengineering, University of California, San Francisco, San Francisco, United States; [2]Department of Orthopaedic Surgery, University of California, San Francisco, San Francisco, United States; [3]National High Magnetic Field Laboratory,Department of Biological Science, Florida State University, Tallahassee, United States; [4]Mass Spectrometry Facility, Department of Pharmaceutical Chemistry, University of California, San Francisco, San Francisco, United States; [5]Department of Bioengineering and Therapeutic Sciences, Department of Otolaryngology–Head and Neck Surgery, Eli and Edythe Broad Center of Regeneration Medicine and Stem Cell Research, University of California, San Francisco, San Francisco, United States

**Abstract** Cell surface receptors are central to the cell's ability to generate coordinated responses to the multitude of biochemical and physical cues in the microenvironment. However, the mechanisms by which receptors enable this concerted cellular response remain unclear. To investigate the effect of cellular tension on cell surface receptors, we combined novel high-resolution imaging and single particle tracking with established biochemical assays to examine TGFβ signaling. We find that TGFβ receptors are discretely organized to segregated spatial domains at the cell surface. Integrin-rich focal adhesions organize TβRII around TβRI, limiting the integration of TβRII while sequestering TβRI at these sites. Disruption of cellular tension leads to a collapse of this spatial organization and drives formation of heteromeric TβRI/TβRII complexes and Smad activation. This work details a novel mechanism by which cellular tension regulates TGFβ receptor organization, multimerization, and function, providing new insight into the mechanisms that integrate biochemical and physical cues.

**\*For correspondence:** tamara. alliston@ucsf.edu

[†]These authors contributed equally to this work

**Competing interests:** The authors declare that no competing interests exist.

## Introduction

The diversity and specificity of cellular responses rely on the precise integration of biochemical and physical cues from the microenvironment. Cells generate a coordinated response through interactions among signaling pathways – from ligands and receptors to intracellular effectors. Receptors are a particularly versatile locus of control since they undergo regulated microdomain clustering, internalization and homo/hetero-meric multimerization. Because these mechanisms affect ligand binding, enzymatic activity, and effector recruitment, receptors play a crucial role in defining signal intensity, duration, location, and quality (*Bethani et al., 2010*; *Di Guglielmo et al., 2003*; *Groves and Kuriyan, 2010*; *Salaita et al., 2010*). However, many questions remain about the

**eLife digest** Cells constantly encounter diverse physical and biological signals in their surroundings. Information contained in these signals is transmitted from the cell surface to the interior to trigger coordinated changes in the cell's behavior. Physical signals include the forces generated by cells pulling on one another or on their surroundings. These pulling forces calibrate the cell's response to biological signals through mechanisms that remain unclear.

The cell surface contains many different proteins that are specialized to sense these signals and guide the cell's response. In animals, these membrane proteins include the receptors that detect a small signaling protein known as TGFβ. TGFβ first binds to one of these receptors (called TβRII). Next another receptor (called TβRI) is recruited to the complex. Once this complex is formed, the TGFβ receptors activate a complicated signaling pathway that controls how cells grow and divide. Previous work has shown that the TGFβ pathway can also sense and respond to mechanical forces. But it remains poorly understood how pulling forces (or tension) impact TGFβ receptors at the cell surface.

Rys, DuFort et al. have now used cutting-edge microscopy and biochemical techniques to analyze individual TβRI and TβRII receptors and observe how they respond to mechanical forces in real-time. This revealed that TβRI and TβRII exist in discrete regions on the cell surface. Rys, DuFort et al. observed that TβRI is enriched at assemblies of molecules called focal adhesions. Focal adhesions are the sites on cell surfaces that allow cells to adhere to one another and to the molecular scaffolding in their surroundings. Unlike TβRI, TβRII was often excluded from these sites and more commonly appeared to 'bounce' around the edges of individual focal adhesions. Therefore, focal adhesions limit the interactions between TβRI and TβRII, by sequestering one away from the other.

Rys, DuFort et al. next treated cells with a chemical that disrupts tension, and saw that the physical separation between TβRI and TβRII collapsed, which permitted these two receptors to interact and form a working signaling complex. Further work is needed to understand how physical control of TGFβ receptor interactions helps cells coordinate their tasks in response to the myriad biological and physical signals in their surroundings.

mechanisms by which receptors participate in the concerted cellular response to a multitude of concurrent cues.

The TGFβ signaling pathway exemplifies the importance of regulated receptor multimerization. TGFβ signals through a heterotetrameric complex of transmembrane receptor kinases. Once the TGFβ ligand is activated from its latent form, it binds directly to a dimer of type II receptors (TβRII) (*Annes, 2003*; *Munger et al., 1999*; *Wipff et al., 2007*; *Munger and Sheppard, 2011*). The ligand-bound TβRII complex recruits and phosphorylates two type I receptors (TβRI) – either Alk5 or Alk1 (*Wrana et al., 2008*). TβRI, in turn, phosphorylates and activates canonical Smad proteins and multiple non-canonical effectors, such as RhoA, TAK1 and Akt (*Massague, 1998*; *Feng and Derynck, 2005*). Specifically, recruitment of Alk5 to the TβRII complex stimulates phosphorylation of Smad2/3, whereas Alk1 recruitment drives activation of Smad1/5/8 (*Lin et al., 2008*). The inappropriate shift of TβRII multimerization partner from Alk5 to Alk1 underlies disease processes ranging from vascular disorders to osteoarthritis (*Blaney Davidson et al., 2009*; *Goumans, 2002*). Not only do TGFβ receptors associate with one another, but also with a number of other receptor families, notably integrins (*Scaffidi et al., 2004*; *Garamszegi et al., 2010*). Garamszegi et al. revealed a physical interaction between integrin α2β1 and TGFβ receptors involved in collagen-induced Smad phosphorylation (*Garamszegi et al., 2010*). TGFβ receptor interactions alter ligand specificity and effector selection, offering a regulatory mechanism to calibrate TGFβ signaling based on the cellular microenvironment.

Integrins, another class of multimeric receptors, are central to cellular mechanotransduction. Upon integrin binding to the extracellular matrix, the formation of focal adhesions stimulates actomyosin contractility to generate cellular tension (*DuFort et al., 2011*; *Giancotti, 1999*; *Ingber, 1997*). Through this Rho/ROCK-dependent mechanism, cells establish tensional homeostasis with the physical features of the extracellular environment (*DuFort et al., 2011*). Cellular tension can amplify, alter, or suppress cellular responses to growth factor signaling (*Allen et al., 2012*;

*McBeath et al., 2004*; *Wang et al., 2012*). The functional state of many intracellular effectors, including β-catenin, YAP/TAZ, and MAPK, is modulated by cellular tension (*Samuel et al., 2011*; *Dupont et al., 2011*; *Wang et al., 1998*). In the case of TGFβ signaling, we and others have identified several mechanosensitive responses (*Allen et al., 2012*; *Wang et al., 2012*; *Leight et al., 2012*). The activation of latent TGFβ ligand, as well as the phosphorylation, nuclear translocation and transactivation of Smads is regulated by cellular tension in a Rho/ROCK-dependent manner (*Allen et al., 2012*; *Wipff and Hinz, 2008*). However, the mechanisms by which changes in cellular tension modulate effector activity remain unclear.

The effect of cellular tension on the multimerization of receptors other than integrins is largely unexplored. In spite of the established tension-sensitive regulation of downstream signaling effectors, the effect of physical cues on growth factor receptor interactions is unknown. This gap in understanding is partly due to the fact that until recently, studies of cell surface receptor colocalization and physical interactions have mostly utilized biochemical, biophysical, or fluorescence imaging approaches. While invaluable, these approaches are limited by their inability to discriminate spatially discrete molecular interactions that occur in specific cellular domains. Novel super-resolution imaging approaches provide the capability to visualize receptor responses to biochemical and physical cues at the single molecule level with spatial and temporal specificity (*Coelho et al., 2013*; *Manley et al., 2008*; *Rossier et al., 2012*; *Calebiro et al., 2013*; *Xia et al., 2013*). To elucidate mechanisms by which physical cues regulate growth factor signaling, we utilize high-resolution imaging, single particle tracking, mass spectrometry and biochemical assays to test the hypothesis that cellular tension regulates TGFβ receptor multimerization. We find that cellular tension controls the spatial organization, multimerization and activity of a discrete population of TGFβ receptors at integrin-rich focal adhesions, suggesting a novel mechanism by which physical cues calibrate the activity of the TGFβ signaling pathway.

## Results

### Discrete localization of TβRI and TβRII to segregated spatial domains

To investigate the spatiotemporal control of TGFβ receptors, we evaluated the localization of endogenous and fluorescently tagged TβRII and TβRI in ATDC5 chondroprogenitor cells and NIH3T3 fibroblasts. Immunofluorescence of TβRII in both wildtype and transfected ATDC5 cells yielded similar results, revealing specific punctate staining that did not provide structural information (*Figure 1A*, *Figure 1—figure supplement 1*). Proceeding with fluorescently tagged TβRII allowed for visualization of fine structural features in static and dynamic conditions. Spinning disc confocal microscopy of TβRII-mEmerald allowed visualization of its spatial organization, revealing shadowed regions where TβRII expression is completely absent (indicated by arrows, *Figure 1B–D*). Total internal reflection fluorescence (TIRF) microscopy improves visualization of transmembrane proteins by examining a thin section of the sample at the adherent cell surface. Switching from widefield microscopy (*Figure 1E*) to TIRF on the same cell vividly revealed segregated domains of TβRII (*Figure 1F*) and TβRI (*Figure 1G,H*). The sequestration of TβRII from TβRI was present with either the canonical (Alk5) or non-canonical (Alk1) type I TGFβ receptors (*Figure 1G,H*). Indeed, when co-expressed in the same cell, TβRII is enriched at the boundary of discrete TβRI domains, demonstrating a novel spatial segregation of these signaling partners (*Figure 1I–L*).

### Single molecule trajectories reveal specific regulation of TGFβ receptor dynamics

Since dynamic recruitment of TβRI to TGFβ-bound TβRII complexes stimulates downstream effectors, we sought to determine if spatial segregation of TGFβ receptors affects receptor mobility. Single-particle tracking photoactivated localization microscopy (sptPALM) resolves the dynamics of individual molecules in live single cells. Using sptPALM, we captured thousands of trajectories of individual TβRI (Alk5) and TβRII proteins labeled with photoswitchable mEos2 (*Figure 2A,B*) (*McKinney et al., 2009*). The large number of long duration molecular trajectories (*Figure 2C*) allowed us to visualize single molecule track behavior and describe molecular environments within individual cells. For both TβRI and TβRII, individual receptors showed a range of mobility, resulting in groups of immobile, confined, or freely diffusive receptors (representative tracks, *Figure 2D*).

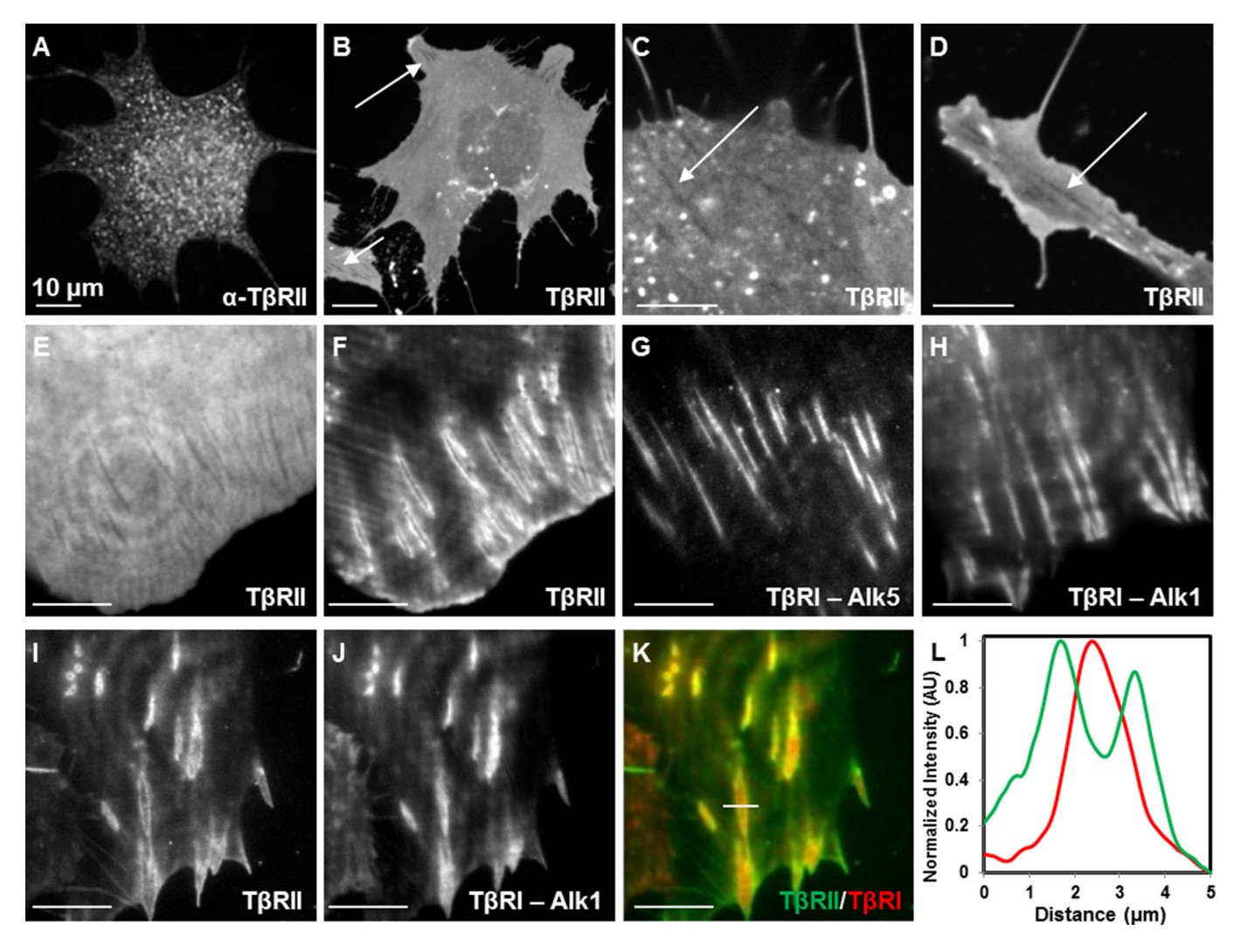

**Figure 1.** Spatial segregation of TβRII from TβRI. Spinning disc confocal imaging of endogenous TβRII (A, *Figure 1—figure supplement 1*) demonstrates punctate staining. Imaging of mEmerald-labeled TβRII (B) reveals TβRII-absent domains in ATDC5 (B,C) and NIH3T3 (D) cells expressing mEmerald-TβRII. Switching from widefield (E) to TIRF mode imaging (F) on the same cell unveils a specific spatial organization of TβRII, which is discrete from that of TβRI (Alk5 and Alk1) (G,H). ATDC5 cells co-expressing mEmerald-TβRII and mCherry-TβRI (Alk1) reveal that TβRII surrounds specific domains of TβRI (I-L). Quantitative profile plot of expression intensity demonstrates separate and distinct localization patterns of TβRI and TβRII (L).

The following figure supplement is available for figure 1:

**Figure supplement 1.** Endogenous staining of TβRII insufficient for spatial organization visualization.

Mobility of each group of TβRI did not differ significantly from TβRII (*Figure 2E*), but the diffusion coefficient of TβRI was slightly higher (*Figure 2F*), perhaps because of its lower molecular weight (TβRI/Alk5 56 kDa vs. TβRII 65 kDa). Relative to whole cell TGFβ receptor dynamics, TβRI and TβRII are significantly less mobile in cellular domains enriched with clusters of spatially organized receptors (*Figure 2F*). Thus, this spatially organized population of TGFβ receptors is slower and more confined, possibly due to interactions with other proteins.

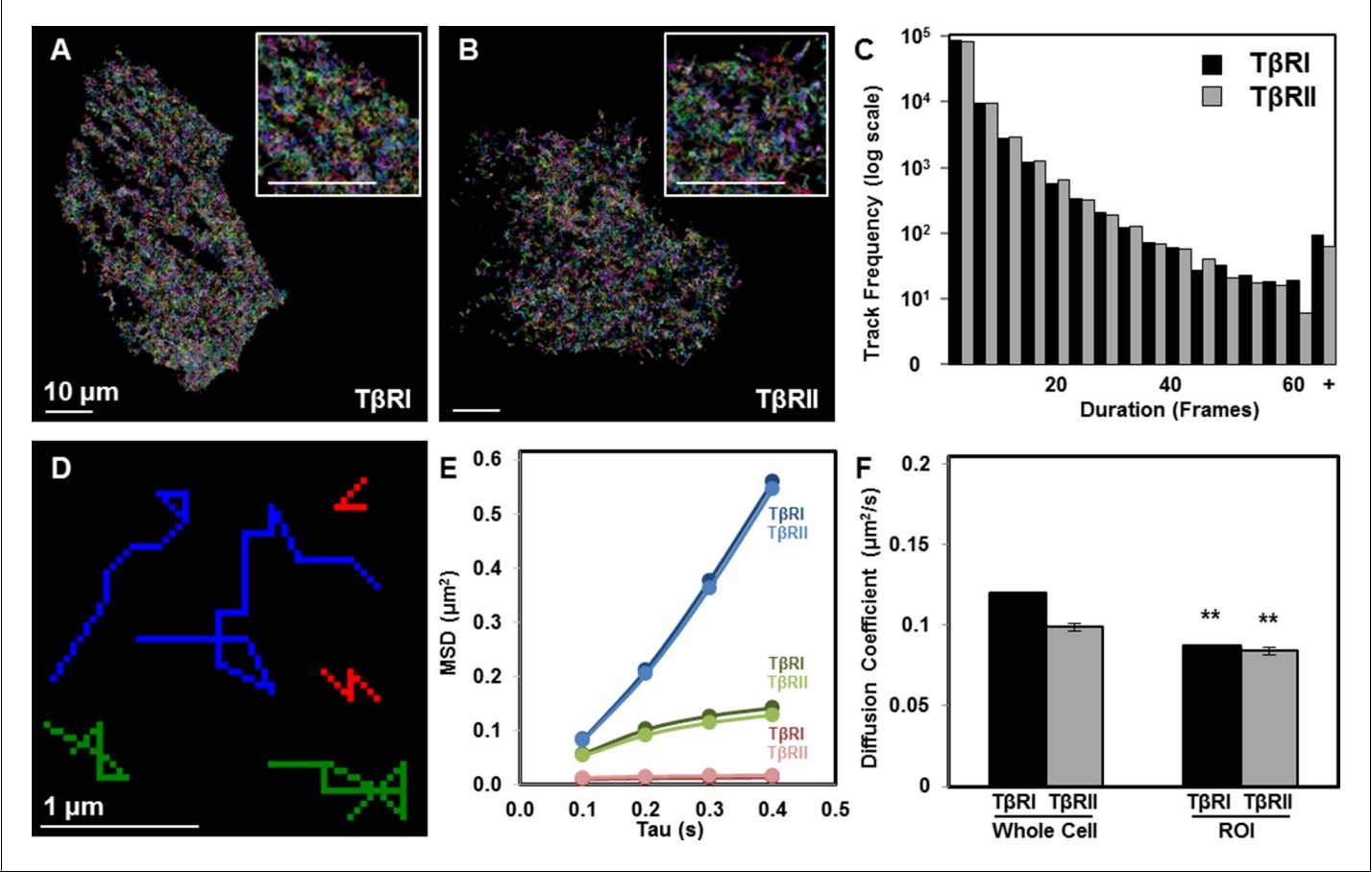

**Figure 2.** Limited TβRI (Alk5) and TβRII mobility in areas of receptor spatial organization. All mEos2-tagged TβRI and TβRII sptPALM single molecule trajectories with durations of at least 5 frames (500 ms) are plotted for representative ATDC5 cells, in which each color represents a different track (**A,B**). Cellular domains outside the imaging plane appear black. The histogram represents the distribution within a single cell of trajectory durations for individual TβRI and TβRII molecules (**C**). Representative individual TβRI sptPALM single molecule trajectories exhibiting immobile (red), confined (green), and freely diffusive (blue) movement are plotted in (**D**), with calculated mean squared displacement (MSD) plots for each population of TβRI and TβRII shown in E (mean ± SEM). Comparison of diffusion coefficients for TβRI and TβRII (F, mean ± SEM) in whole cells relative to areas of segregated TβRI/TβRII identify a less mobile population of TGFβ receptors in these regions of interest (ROI). See **Source code 1** and **Figure 2—source data 1**.

The following source data is available for figure 2:

**Source data 1.** sptPALM single molecule trajectories

## Focal adhesions organize TβRII around a segregated pool of TβRI

The distinct localization of TGFβ receptors could result from physical interactions with any number of known TGFβ receptor-associated proteins. Among these, integrins bind to TβRI and TβRII and functionally interact with the TGFβ pathway at multiple levels (*Wrana et al., 2008*; *Scaffidi et al., 2004*). The primary integrins in chondrocytes are integrins α2 and αV, which bind collagen and vitronectin/fibronectin (*Loeser, 2000*). Both integrins interact with the TGFβ pathway (*Scaffidi et al., 2004*; *Garamszegi et al., 2010*). TIRF imaging of mCherry-labelled integrin α2 revealed the presence of focal adhesions at these TβR-rich sites. Specifically, TβRII is absent from sites of adhesions and forms a peripheral ring surrounding integrin α2, resulting in distinct patterns of spatial localization (*Figure 3A*). Interestingly, this spatial organization is absent in cells grown on poly-l-lysine-coated substrates that facilitate integrin-independent cell adhesion (*Figure 3—figure supplement 1*). Therefore, TβRII organization at sites of adhesion is dependent upon integrin activity. Profile plots of intensity and a custom analysis (*Figure 3Ai,Bi,Ci*) were utilized to quantify colocalization between

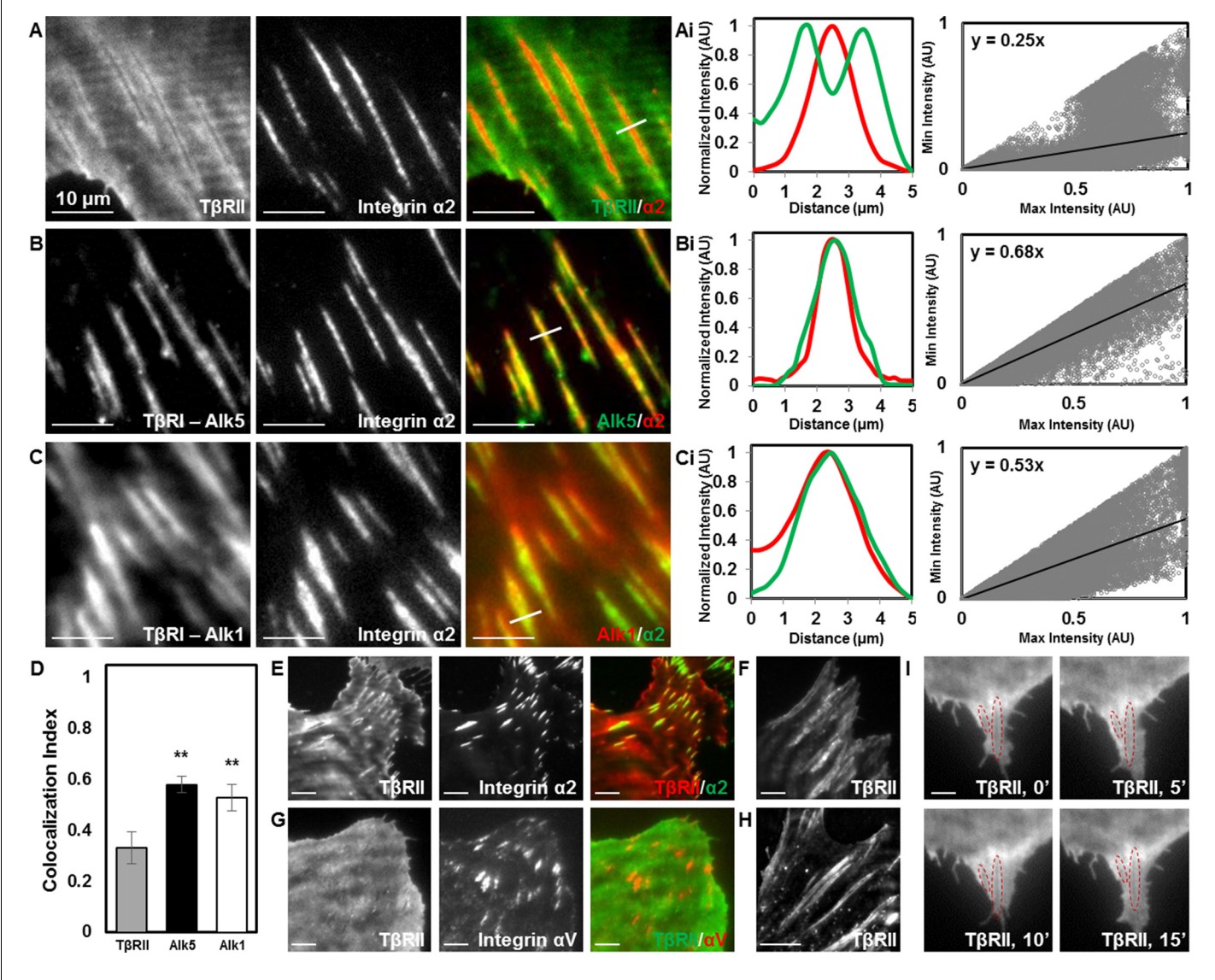

**Figure 3.** Focal adhesions sequester TβRI from TβRII. TIRF mode imaging and a custom colocalization analysis were used to evaluate localization of TβRII (**A**), Alk5 (**B**), or Alk1 (**C**) with integrin α2 in ATDC5 cells. TβRII surrounds integrin α2 (**A**), whereas both subtypes of TβRI, Alk5 (**B**) and Alk1 (**C**), are included within integrin-rich focal adhesions, as reflected by profile plots and the slope values of the regression lines (Ai,Bi,Ci). Quantification of colocalization reveals that Alk5 and Alk1 are significantly more colocalized with integrin α2 relative to TβRII (**p < 0.001, mean ± SD, D, *Figure 3— source data 1*). This organization is also present in ATDC5 cells when the fluorescent labels for TβRII and integrin α2 have been switched (**E**), in osteosarcoma Saos-2 cells (**F**), or in epithelial MCF10A cells (**G**), when labeling focal adhesions with integrin αV (**G**), and when TβRII is expressed and imaged alone (**H**). TβRII spatial organization is unaffected by addition of TGFβ, indicated by red outlines in the same cellular region following 15 min of TGFβ treatment (**I**). See *Source code 2*.

The following source data and figure supplement are available for figure 3:

**Source data 1.** Colocalization Index
**Figure supplement 1.** Focal adhesion formation and TβRII spatial organization are dependent on integrin activity.

TβRs and integrin α2 across multiple cells. The slope of the regression line can be used as a metric, in which higher values indicate increased colocalization of two proteins. TβRI (Alk5 and Alk1) is precisely colocalized with integrin α2 within focal adhesions, such that adhesions appear yellow (*Figure 3B,C*) and regression line slopes (*Figure 3Bi,Ci*) are higher relative to TβRII (*Figure 3A,Ai*).

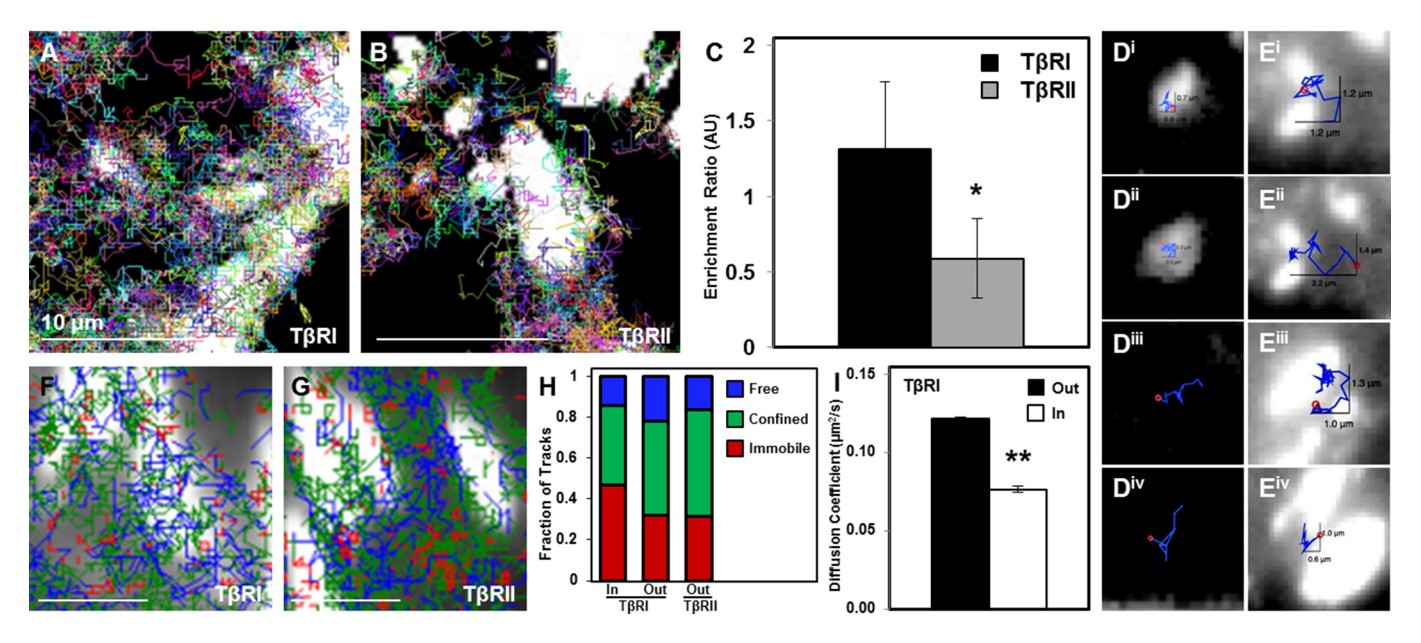

**Figure 4.** Dynamic interaction of TβRs with integrins facilitate spatial organization. Representative trajectories for TβRI (Alk5) overlaid with the tagged focal adhesion marker vinculin are consistent with TIRF results showing a colocalization and interaction between integrin-based adhesions and TβRI (A) but not TβRII (B). Quantification of these regions shows that TβRI is preferentially enriched inside adhesions relative to outside, and that TβRII is preferentially excluded at these same sites (*p < 0.01, mean ± SD, C). Representative single molecule trajectories show sequestration of TβRI in focal adhesions (D, i-ii) and free diffusion outside adhesions (D, iii-iv), whereas TβRII bounces around the edges of focal adhesions in a freely diffusive (E, i-ii) or confined (E, iii-iv) manner. Analyzing TβR trajectories at focal adhesions based on diffusion (Red: Immobile, Green: Confined, Blue: Freely Diffusive) shows a higher density of tracks inside adhesions for TβRI (F) compared to TβRII (G), and demonstrates a higher fraction of immobile TβRI tracks inside relative to outside adhesions (H). The diffusion coefficient of TβRI trajectories decreases inside adhesions (mean ± SD, I). See *Source code 1* and *Figure 4—source data 1*.

The following source data is available for figure 4:

**Source data 1.** Enrichment Ratio and Diffusion Coefficient

This analysis reveals that integrin α2 colocalizes significantly more with Alk5 and Alk1 than with TβRII (*Figure 3D*). The specific localization of TβRII near focal adhesions is apparent in cells of both mesenchymal (ATDC5, *Figure 3A–C,E*; Saos-2, *Figure 3F*) and epithelial (MCF10A, *Figure 3G*) origin and is observed whether integrin α2 or integrin αV is tagged with a fluorescent protein (*Figure 3*). Furthermore, this observation still holds if the fluorescent labels for TβRII and integrin α2 are switched, as well as if TβRII is expressed and imaged alone (*Figure 3E,H*). The overall spatial organization of TGFβ receptors at sites of adhesion is not affected upon stimulation with exogenous TGFβ (*Figure 3I*), suggesting that this spatiotemporal organization is regulated through mechanisms independent from TGFβ ligand addition. Given the critical role of integrins in mechanotransduction and the known sensitivity of TGFβ signaling to cellular tension (*Allen et al., 2012*), the unique pattern of TGFβ receptor and integrin localization could prime TGFβ receptors for regulation by elements of the mechanotransduction pathway.

### Focal adhesions immobilize TβRI and limit the integration of TβRII

To investigate the effect of focal adhesions on TβRI (Alk5) and TβRII dynamics, we used sptPALM to visualize TGFβ receptor trajectories near or within these vinculin-rich domains (*Figure 4A,B*). SptPALM shows, both qualitatively and quantitatively, that TβRI is preferentially enriched and TβRII is preferentially excluded at sites of adhesion (*Figure 4A–C*). Analysis of individual TβRII trajectories shows that TβRII 'bounces' around the edges of individual focal adhesions (*Figure 4E*) but is rarely incorporated within the focal adhesion, as is common for TβRI (*Figure 4D,i–ii*). To determine if focal adhesions shifted the fractions of freely diffusive, confined, or immobile receptors, TβRI and TβRII

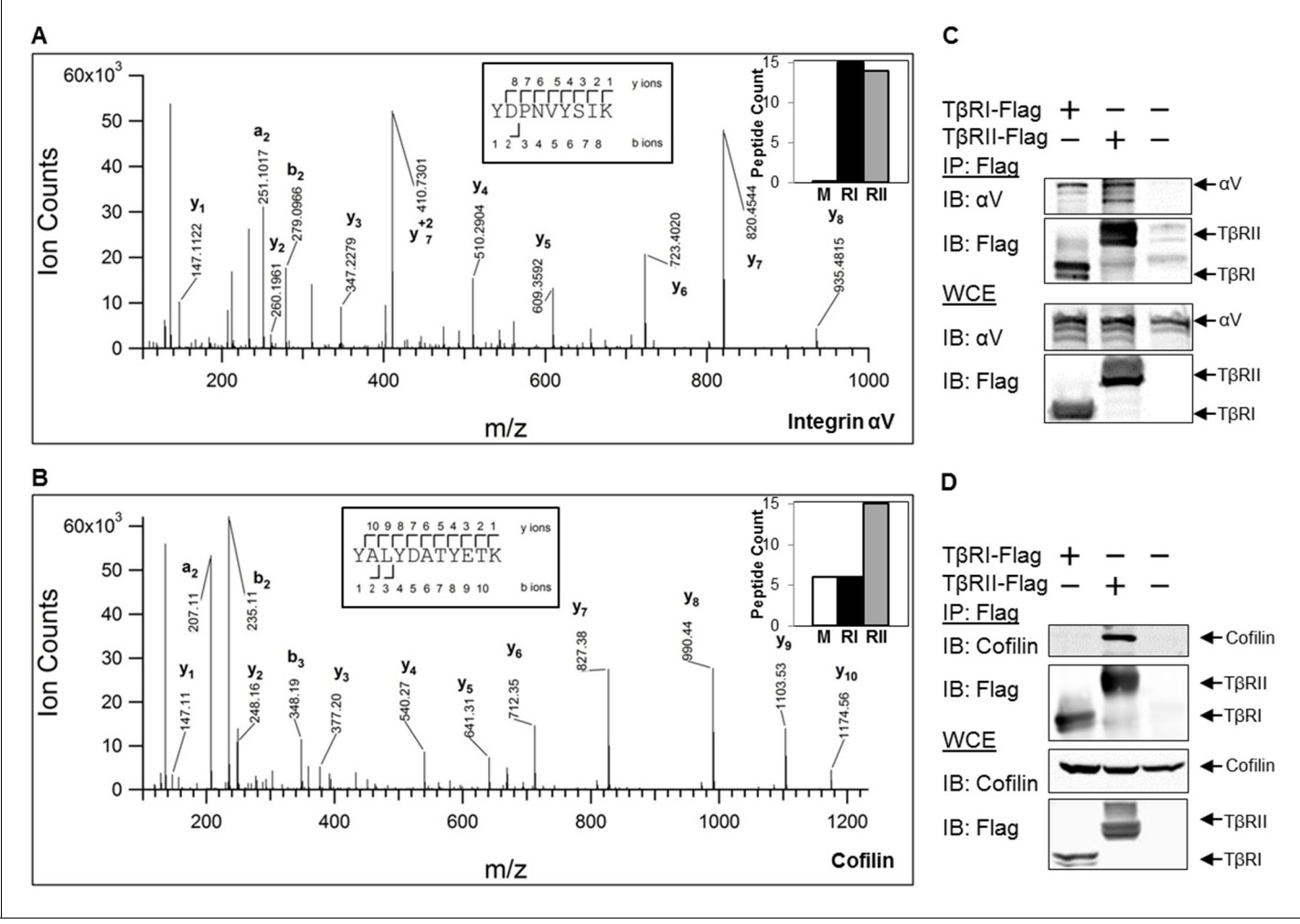

**Figure 5.** TβRs form complexes with integrin αV and cofilin. High-energy collision dissociation–tandem mass spectra obtained from precursor ions with mass 549.7775+2 (**A**) and 669.3185+2 (**B**) found in tryptic digests of immunoaffinity pulldowns of TβRI/II, corresponding to peptides spanning residues Y153-K165 of human integrin αV (**A**) and Y82-K92 of human cofilin (**B**). b- and y- type ion series are labeled in the figure. Insets show the sequences of the peptides as well as representative peptide counts for integrin αV (**A**) and cofilin (**B**) for mock (**M**), TβRI (RI, Alk5), and TβRII (RII) pulldowns. Co-immunoprecipitation of Flag-tagged TβRI and TβRII demonstrate the presence of integrin αV and cofilin in these complexes (**C,D**).

trajectories near sites of adhesion were mapped based on receptor mobility. Trajectory maps reveal that TβRII mobility is confined near focal adhesions, which sequester and immobilize TβRI (*Figure 4F,G*). Indeed, a higher fraction of immobilized TβRI is present inside adhesions relative to outside (*Figure 4H*). Accordingly, the diffusion coefficient for TβRI decreases for tracks inside adhesions compared to those outside, demonstrating that this spatial organization specifically limits TβRI mobility (*Figure 4I*). The differential localization and dynamics of TβRI and TβRII in adhesion-rich domains, relative to one another and to the whole cell TGFβ receptor population, indicates that this spatial control has functional implications for TGFβ signaling and for mechanotransduction.

## TGFβ receptors form complexes with integrin αV and the actin-binding protein cofilin

To determine whether these changes in receptor mobility at sites of adhesion are due to direct or indirect physical interactions with other proteins, we performed mass spectrometry and co-immunoprecipitation experiments. Mass spectrometric analysis of proteins that precipitate with Flag-tagged TβRI (Alk5) and TβRII revealed hundreds of proteins, several of which were specifically enriched compared with precipitates of untransfected (mock) cells. The analysis identified proteins already known

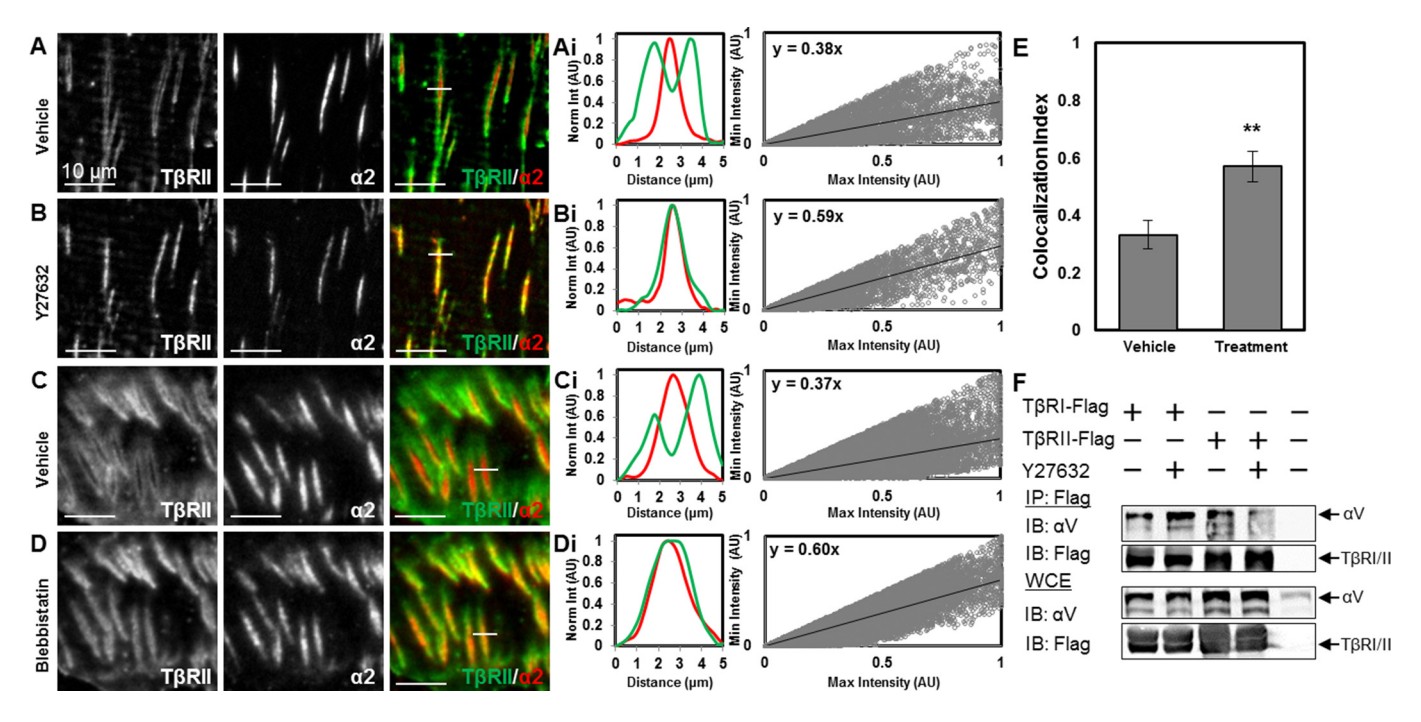

**Figure 6.** Tension-sensitive regulation of TβR spatial organization. Within 15 min of disrupting cellular tension by adding the ROCK inhibitor Y27632 (**A,** **B**) or the myosin II inhibitor blebbistatin (**C,D**), the peripheral ring of TβRII-mEmerald around focal adhesions (**A,C**) completely collapses (**B,D**). Colocalization quantification (**Ai,Bi,Ci**) demonstrates that TβRII is significantly more colocalized with integrin α2 post-treatment (Y27632, blebbistatin) relative to pre-treatment (**\*\*p < 0.001, mean ± SD, E, *Figure 6—source data 1***). Disruption of tension with Y27632 enhances integrin αV association with TβRI but reduces its association with TβRII (**F**). See *Source code 2*.

The following source data is available for figure 6:

**Source data 1.** Colocalization Index (vehicle and treatment)

to interact with TβRs, such as PRMT5 and PRMT1 (*Xu et al., 2013*). TβRs also precipitated several adhesion-related proteins, including integrin αV and endogenous cofilin, as shown in the annotated spectra (*Figure 5A,B*). The peptide counts (graph insets) indicate that integrin αV associates with both TβRI and TβRII, and that cofilin preferentially associates with TβRII (*Figure 5A,B*). Cofilin is an actin-binding protein that severs ADP-actin filaments at the leading edge of migratory cells (*Pollard and Borisy, 2003*). Previous reports implicate cofilin as a target of TGFβ-activated RhoA, which promotes actin reorganization through ROCK, LIMK and cofilin (*Vardouli et al., 2005*; *Lamouille et al., 2014*). However, this is the first report, to our knowledge, of a complex between TGFβ receptors and cofilin. To confirm these mass spectrometry findings, we performed co-immuno-precipitation on cells expressing Flag-tagged TβRI/II and tagged integrin αV or cofilin (*Figure 5C,D*). Consistent with the mass spectrometry peptide counts, integrin αV forms a complex with both TβRI and TβRII, whereas cofilin primarily interacts with TβRII. Although the novel finding of a complex formation, either through direct or indirect interactions, between TβRII and cofilin remains to be further explored, it suggests a potential mechanism underlying the discrete spatial organization of TβRII at focal adhesions.

## Cellular tension regulates TGFβ receptor organization at focal adhesions

Integrins transmit changes in the physical microenvironment across the plasma membrane to modulate cellular tension and signaling. The presence of a focal adhesion-associated TGFβ-receptor population suggests a novel mechanism by which cellular tension may regulate TGFβ signaling. To test the hypothesis that TGFβ receptor organization at focal adhesions is sensitive to cellular tension, we

treated ATDC5 cells with the ROCK inhibitor Y27632 or the myosin II inhibitor blebbistatin. Within 15 min of adding Y27632 (*Figure 6A,B*) or blebbistatin (*Figure 6C,D*), the peripheral ring of TβRII completely collapses. The segregation of TβRII from TβRI and integrin α2 at sites of adhesion is dynamically released, such that TβRII (*Video 1*) converges and colocalizes with integrin α2 (*Video 2*, *Video 3*). Quantitative analysis demonstrates that TβRII is significantly more colocalized with integrin α2 after addition of Y27632 and blebbistatin (*Figure 6E*).

To assess the effect of cellular tension on physical associations among TβRI, TβRII and integrins, we performed co-immunoprecipitation experiments. We find that cellular tension not only regulates the spatial organization of integrins and TGFβ receptors, but also affects their physical associations with each other; though these interactions may be direct or indirect. Specifically, while disruption of tension with the ROCK inhibitor enhanced integrin αV association with TβRI, it almost completely blocked the association between integrin αV and TβRII (*Figure 6F*).

## Tension-sensitive regulation of TGFβ receptor heteromerization and signaling

Since a reduction in cellular tension drives colocalization of TβRI and TβRII, we sought to determine if this change in spatial organization had functional consequences for TGFβ signaling. We first evaluated the effect of reduced cellular tension on TβRI/TβRII heteromerization using co-immunoprecipitation. Release of this discrete spatial segregation of TGFβ receptors at focal adhesions allows the receptor subunits to interact such that ROCK-inhibition stimulates formation of heteromeric TβRI/TβRII complexes (*Figure 7A*). To examine the effect of manipulating cellular tension under physiological conditions, we cultured cells on polydimethylsiloxane (PDMS) substrates of varying stiffness. A reduction in cellular tension through culture on compliant substrates significantly drives TβRI/TβRII complex formation (*Figure 7B*). Therefore, a reduction in cellular tension, due to pharmacologic ROCK inhibition or changes to the stiffness of the microenvironment, drives formation of a multimeric TβRI/TβRII complex that is required for the activation of downstream TGFβ effectors.

To determine the effect of tension-sensitive TβR localization and heteromerization on downstream TGFβ effectors, we evaluated the phosphorylation of Smad3. Culturing cells on compliant substrates leads to significantly increased endogenous Smad3 phosphorylation (*Figure 7C*). Interestingly, the effect of TGFβ on Smad3 phosphorylation is substrate-dependent, such that TGFβ induces Smad3 phosphorylation on 0.5 kPa substrates but not on 16 kPa substrates (*Figure 7C*). This is consistent with the established non-linear response of TGFβ signaling and other cellular behaviors to cellular tension (*Allen et al., 2012*; *Rape et al., 2015*). Thus the spatial organization of TβRI and

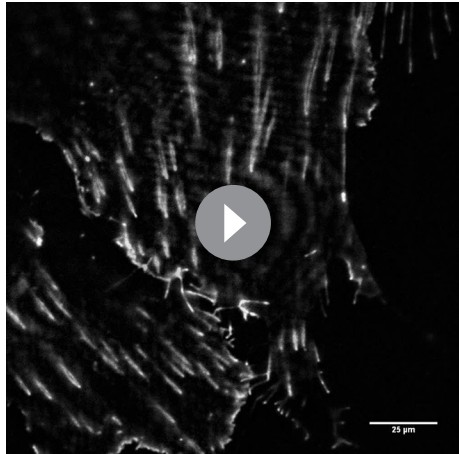

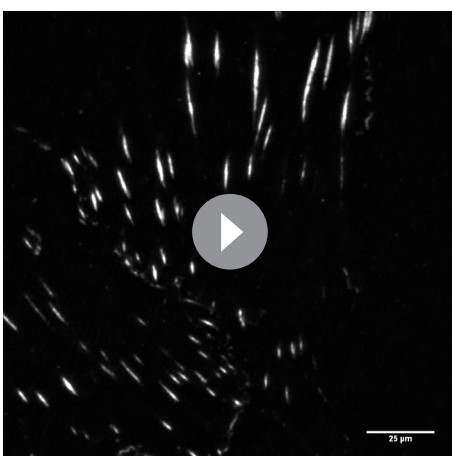

**Video 1.** Disruption of cellular tension leads to dynamic disassembly of TβRII spatial organization at sites of adhesion (*Figure 6*). TβRII-mEmerald spatial organization collapses within 15 min of adding ROCK inhibitor Y27632 in ATDC5 cells (45 min, 7 fps).

**Video 2.** Disruption of cellular tension leads to dynamic disassembly of TβRII spatial organization at sites of adhesion (*Figure 6*). Integrin α2-mCherry adhesions disassemble within 15 min of adding ROCK inhibitor Y27632 in ATDC5 cells (45 min, 7 fps).

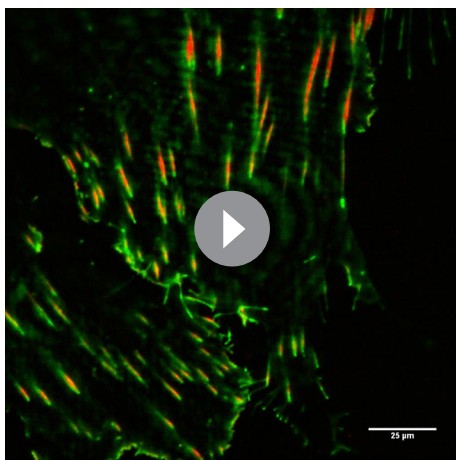

**Video 3.** Disruption of cellular tension leads to dynamic disassembly of TβRII spatial organization at sites of adhesion (*Figure 6*). Composite of TβRII and integrin α2 (*Video 3*) demonstrate a tension-sensitive collapse of this discrete spatial organization at sites of adhesion and a reorganization at the cell periphery.

TβRII by integrins at focal adhesions affords tension-sensitive control of TβRI and TβRII multimerization and activation of Smad3, providing a mechanosensitive mechanism by which cells calibrate their response to TGFβ.

## Discussion

Here we show that cellular tension regulates TGFβ receptor spatial organization and interactions at focal adhesions, providing a novel mechanism for the cellular integration of signaling by physical and biochemical cues. We observe a novel spatiotemporal regulation of the TGFβ pathway such that TβRII is segregated from TβRI and integrins at sites of adhesions. Single particle tracking reveals the dynamics of individual TGFβ receptor molecules, and identifies populations of TGFβ receptors with distinct behaviors and mobility near and far from sites of focal adhesions. The confined population of TGFβ receptors at focal adhesions has lower mobility than the freely diffusive receptor population far from sites of adhesion. TGFβ receptors associate with several adhesion-related proteins, including the actin-binding protein cofilin, which preferentially associates with TβRII relative to TβRI. This novel spatial organization of TβRI and TβRII at sites of adhesion provides mechanosensitive control of TGFβ receptor multimerization and function independently of TGFβ ligand stimulation. Overall, this reveals the potential of two differentially regulated populations of TGFβ receptors – one that is TGFβ-sensitive and one that is tension-sensitive – a finding that may contribute to the context-dependent signaling outcomes of this pathway.

This tension-dependent mechanism for the regulation of TGFβ receptors has a number of interesting functional implications. At the level of the TGFβ ligand, integrins activate TGFβ from its latent form through cellular tension generated by actomyosin contraction (*Wipff et al., 2007*; *Munger and Sheppard, 2011*; *Wells and Discher, 2008*; *Giacomini et al., 2012*). The observed recruitment of TGFβ receptors to focal adhesions would enrich their access to this reservoir of integrin-activated TGFβ. At the receptor level, focal adhesions may sequester TβRI from TβRII to limit their activity in the presence of ligand. The extent to which this sequestration is cofilin-dependent requires further investigation. This sequestration of TβRI may contribute to its slow internalization, relative to TβRII, following TGFβ stimulation (*Vizan et al., 2013*; *Ma et al., 2007*). Alternatively, focal adhesions may create structured TβRI and TβRII boundaries that prime a robust response when cells encounter the correct combination of physical and biochemical cues. We demonstrate tension-sensitive regulation of endogenous downstream Smad3 phosphorylation by cellular tension and TGFβ. In addition, chondrocytes grown in TGFβ on 0.5 MPa substrates induce differentiation markers far beyond levels induced by either cue alone (*Allen et al., 2012*). We and others have reported that the effect of substrate stiffness or cellular tension/Rho/ROCK activity on downstream TGFβ signaling is synergistic and nonlinear (*Allen et al., 2012*; *Wang et al., 2012*; *Leight et al., 2012*). Therefore, it is possible that lower cell tension in one cell type may have a differential effect on Smad phosphorylation, nuclear localization, and transactivation than in another cell type. It would be interesting to examine this effect utilizing a substrate system that provides independent and continuous gradients of ligand density and substrate stiffness (*Rape et al., 2015*). The mechanisms responsible for such synergy have been unclear, but this newly described regulation of TGFβ receptor multimerization and downstream signaling may couple the mechanosensitive activity of the TGFβ pathway to physical cues. Fully understanding the functional implications of this spatially-distinct TGFβ receptor population will require the development of new imaging tools, such as those that can dynamically visualize TGFβ effector activity locally at focal adhesions.

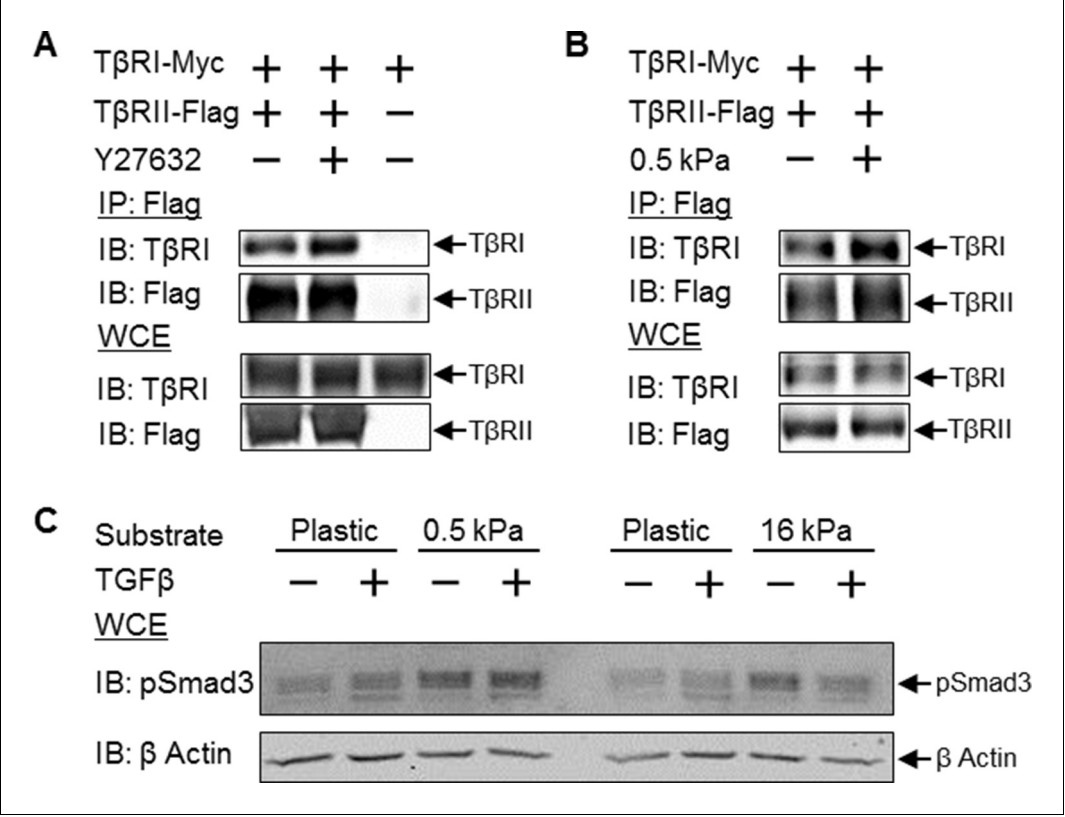

**Figure 7.** Disruption of tension-sensitive TβR segregation increases TβRI/TβRII multimerization and phosphorylation of Smad3. ROCK inhibition releases the discrete spatial organization of TβRs at focal adhesions and drives the formation of heteromeric TβRI/TβRII complexes within 15 min of Y27632 exposure (**A**), as shown by Flag co-immunoprecipitation (IP) and immunoblotting (IB). Likewise, manipulation of cellular tension through culturing cells on collagen II-coated glass or 0.5 kPa PDMS substrates increases co-immunoprecipitation of TβRI with Flag-tagged TβRII ($p < 0.05$, **B**). In cells grown on collagen II-coated compliant (0.5 kPa, $p < 0.05$) or stiff (16 kPa) PDMS substrates, endogenous Smad3 phosphorylation is increased (**C**). The effect of TGFβ on Smad3 phosphorylation is substrate-dependent, such that maximal TGFβ-inducibility is observed on 0.5 kPa substrates ($p < 0.05$), consistent with a tension-sensitive calibration of TβR localization and activity (**C**). See *Figure 7 – source data.*

The following source data is available for figure 7:

**Source data 1.** Western Quantitative Analysis

The current study of TGFβ receptors opens the possibility that tension-sensitive receptor multi-merization may underlie mechanosensitive signaling by other pathways. Cellular tension impacts the activation, translocation, and function of intracellular effectors including small GTP-ases, kinases and transcriptional regulators such as Smads and YAP/TAZ (*Allen et al., 2012*; *Wang et al., 2012*; *Dupont et al., 2011*; *Leight et al., 2012*). However, known mechanisms are insufficient to explain the ability of physical cues to modulate cell-type specific responses to BMP, EGF, and other growth factors (*Wang et al., 2012*; *Paszek et al., 2005*). Several receptor families share features with TGFβ receptors that may contribute to their mechanosensitivity, such as their association with integrin-rich focal adhesions and their potential for the formation of stable receptor clusters by geometric con-straints (*Bethani et al., 2010*; *Salaita et al., 2010*; *Hartman and Groves, 2011*). Previous studies have established important physical and functional links between focal adhesion components and growth factor receptors. EGF receptor binds actin and colocalizes with integrin α2β1 (*Alam et al., 2007*), while the receptor CD44 interacts with several components of the focal adhesion complex, such that hyaluronan-bound CD44 activates c-Src and Rac1 (*Turley et al., 2002*). Aside from TGFβ, shown herein, the extent to which these physical associations contribute to mechanosensitive control

of receptor multimerization or downstream signaling remains to be determined. Nonetheless, others have postulated receptor multimerization as a mechanism for mechanocoupling of TGFβ, ephrin, and T cell receptor signaling (*Hartman and Groves, 2011*; *Hynes, 2009*). In each case, the solid state presentation of the ligand is thought to play a critical role in structuring multimeric receptor clusters. In T cell receptor and ephrin signaling, the solid state is provided by ligands on the neighboring cell, which create geometric constraints that mechanically trap receptors to induce clustering (*Bethani et al., 2010*; *Salaita et al., 2010*; *Hartman and Groves, 2011*). For growth factors like TGFβ, BMP, and EGF, the ECM serves as the solid state (*Hynes, 2009*). ECM proteins such as collagen II bind both TGFβ and integrin α2β1 (*Zhu, 1999*), imposing geometric constraints that may structure receptor clusters. Therefore, growth factor receptor multimerization at focal adhesions, controlled by receptor interactions with integrins and with solid state growth factors, provide focal adhesions with the capability to integrate signaling between physical and biochemical cues.

Understanding the mechanosensitive regulation of TGFβ signaling has significant biological implications. We find that focal adhesions segregate TβRI from TβRII in both epithelial and mesenchymal cell lineages, opening the possibility that this is a general cellular mechanism for the control of TGFβ signaling. It will be interesting to determine if TGFβ receptor multimerization at focal adhesions responds to physical cues that aberrantly promote TGFβ-induced epithelial-mesenchymal transition (EMT) in cancer or the loss of chondrocyte homeostasis in osteoarthritis. On stiff substrates, TGFβ preferentially activates PI3K to induce EMT instead of apoptosis (*Leight et al., 2012*). In osteoarthritis, the material properties of the cartilage ECM deteriorate as chondrocytes inappropriately shift the balance from canonical (Alk5/Smad2/3) to non-canonical (Alk1/Smad1/5/8) TβRI signaling (*Blaney Davidson et al., 2009*). In each case, the extent to which changing the physical environment alters TGFβ effector selection through differential TGFβ receptor multimerization remains to be determined. Applied physical cues, such as compression or shear flow, also regulate TGFβ signaling in cartilage, vasculature, and other tissues (*Li et al., 2010*; *Sakai et al., 1998*; *Streuli, 1993*). Whether similar mechanisms operate in response to exogenous physical cues remains to be elucidated.

In conclusion, we utilized novel high-resolution imaging and single particle tracking microscopy coupled with biochemical assays to explore the spatial organization of TGFβ signaling at the receptor level. At focal adhesions, TβRII is uniquely segregated from its TβRI counterpart. Cellular tension modulates the spatial organization, multimerization, and downstream signaling of TGFβ receptors at sites of adhesion, suggesting the existence of a functionally distinct subpopulation of TGFβ receptors. Overall, this finding provides a new mechanism by which cellular tension and physical cues exert control of growth factor signaling at the cellular membrane.

## Materials and methods

### Plasmids

The plasmids pRK5 TGFβ type I receptor Flag and pRK5 TGFβ type II receptor Flag were gifts from Rik Derynck (Addgene plasmids 14,831 (*Feng and Derynck, 1996*), 31719). The plasmid pRK5 TGFβ type I receptor Myc was also a gift from Rik Derynck. All fluorescent protein expression vectors are available in the Michael Davidson Fluorescent Protein Collection on Addgene. All fluorescent protein expression vectors were constructed using C1 or N1 (Clontech-style) cloning vectors and initially characterized using the advanced EGFP variant mEmerald to verify proper localization of the fusions. To construct the N-terminal (with respect to the fluorescent protein) human integrin alpha2 (NM_002203.3) fusions, the following primers were used to amplify the integrin alpha 2, and create the 18-amino acid linker (GSAGGSGVPRARDPPVAT):

XhoI forward: **CTC CGT CTC GAG ACC GCC ATG GGG CCA GAA CGG ACA GGG GCC**
KpnI reverse: **CGG AAC GGT ACC CCG CTT CCG CCT GCG CTG CCG CTA CTG AGC TCT GTG GTC TCA TCA ATC TCA TCT GGA TTT TTG GTC**

Following digestion and gel purification, the PCR product was ligated into a similarly digested mEmerald-N1 cloning vector to produce mEmerald-Integrin alpha2-N-18. The resulting fusion, along with mCherry-N1 cloning vector, was sequentially digested with AgeI and NotI to yield mCherry-Integrin alpha2-N-18. To generate the N-terminal human integrin alpha V (NM_002210.4) fusions,

the following primers were used to amplify the protein and create the 25-amino acid linker (PGSRAQASNSAVDGTAGPGSPPVAT):

AgeI forward: **CCC GGG ATC CAC CGG TCG CCA CCA TGG CTT TTC CGC CGC GGC GAC GGC TGC GCC TCG GTC**

HindIII reverse: **AAT TGA AGC TTG AGC TCG AGA TCC CGG AAG TTT CTG AGT TTC CTT CAC CAT TTT CAT GAG GTT GAA GCT GCT CCC TTT CTT GTT CTT CTT GAG**

The PCR product was digested, gel purified, and ligated into a similarly treated mEmerald-N1 or mCherry-N1 cloning vector to yield the mEmerald-Integrin alphaV-25 or mCherry-Integrin-alphaV-25 fusions. To construct the N-terminal tagged human vinculin (NM_003373.3) plasmids, the following primers were used to PCR-amplify and create a 21-amino acid linker (SGGSGILQSTVPRARDPPVAT):

NheI forward: **GTC AGA TCC GCT AGC ACC GCC ACC ATG CCA GTG TTT CAT ACG CGC ACG ATC GAG AGC**

EcoR1 reverse: **CGA CTG CAG AAT TCC GCT GCC ACC GGA CTG GTA CCA GGG AGT CTT TCT AAC CCA GCG CAG**

The PCR product was digested and ligated into a similarly cut mEmerald-N1 or mCherry-N1 cloning vector to yield mEmerald-Vinculin-N-21 or mCherry-Vinculin-N-21 expression vectors. To construct the C-terminal human TβR2 (NM_001024847.2) fusion plasmids, the following primers were used to amplify the TβR2 and generate an 18-amino acid linker (SGLRSRESGSGGSSGSGS):

XhoI forward: **GAC GAG CTC GAG AGA GTG GCT CTG TGG GTC GAG TGG AAG TGG CA GCG GTC GGG GGC TGC TCA GGG GCC TG**

BamHI reverse: **CGT CTA GGA TCC CTA TTT GGT AGT GTT TAG GGA GCC GTC TTC AGG AAT CTT CTC C**

Following digestion and gel purification, the PCR product was ligated into a similarly digested mEmerald-C1 cloning vector, to produce mEmerald-TβRII-C-18. The fusion, along with mCherry-C1 and mEos2-C1 cloning vectors, was sequentially digested with AgeI and BamHI and ligated to yield mCherry-TβRII-C-18 and mEos2-TβRII-C-18. To generate the N-terminal human TβRII plasmids and create an 18-amino acid linker (SSGGASAASGSADPPVAT), the following primers were used:

NheI forward: **CGA TCC GCT AGC GCC ACC ATG GGT CGG GGG CTG CTC AGG GGC**

BamHI reverse: **CCT GTA CGG ATC CGC GCT ACC ACT GGC TGC GCT TGC TCC ACC GCT GCT TTT GGT AGT GTT TAG GGA GCC GTC TTC AGG AAT CTT CTC C**

The PCR fragment was digested, gel purified, and ligated with a similarly treated mEmerald-N1 cloning vector to produce mEmerald-TβRII-N-18. The resulting fusion, along with mCherry-N1 and mEos2-N1, was double digested with BamHI and NotI to yield mCherry-TβRII-N-18 and mEos2-TβRII-N-18 respectively. To construct the N-terminal human Alk1 (NM_000020.2) expression vectors, the following primers were used to amplify the plasmid, and create a 13-amino acid linker (GSAGGSGDPPVAT):

EcoRI forward: **GCG TTG AAT TCA CCG CCA TGA CCT TGG GCT CCC CCA GGA AAG GCC**

BamHI reverse: **CGG AAC GGA TCC CCG CTT CCG CCT GCG CTG CCT TGA ATC ACT TTA GGC TTC TCT GGA CTG TTG CTA ATT TTT TGT AGT GTC TTC TTG ATC**

Following amplification, the PCR fragment was digested, purified, and ligated to a similarly treated mEmerald-N1 cloning vector to yield mEmerald-Alk1-N-13. Upon sequence verification, the resulting fusion, along with mCherry-N1, was digested with BamH1 and NotI and ligated to yield mCherry-Alk1-N-13. To generate the C-terminal human Alk5 (NM_004612.2) expression vectors, the following primers were used to amplify the plasmid, and create an 18-amino acid linker (SGLRSGSSAGSASGGSGS):

BglII forward: **GAC TCG AGA TCT GGC TCC AGC GCA GGC AGC GCA TCC GGC GGA AGC GGA AGC GAG GCG GCG GTC GCT GCT CCG CGT C**

HindIII reverse: **CGG TCA AAG CTT TTA CAT TTT GAT GCC TTC CTG TTG ACT GAG TTG CGA TAA TGT TTT CTT AAT CCG C**

Following amplification, the PCR fragment was digested, purified, and ligated to a similarly treated mEmerald-C1 cloning vector to yield mEmerald-Alk5-C-18. The resulting fusion, along with mCherry-C1 and mEos2-C1, was double digested with BglII and NheI to yield mCherry-Alk5-C-18 and mEos2-Alk5-C-18. To construct the N-terminally labeled Alk5 fusions, the following primers were used to amplify the Alk5 and generate a 13-amino acid linker (GSGGAGGGGPVAT):

BglII forward: **GTC TGT AGA TCT GCC ACC ATG GAG GCG GCG GTC GCT GCT CCG**

AgeI reverse: **CGG TCA ACC GGT CCT CCG CCG CCC GCA CCC CCG GAA CCC ATT TTG ATG CCT TCC TGT TGA CTG AGT TGC GAT AAT GTT TTC TTA ATC CGC**

Following amplification, the resulting fragment was digested, purified, and ligated to a similarly treated mEmerald-N1 cloning vector, resulting in mEmerald-Alk5-N-13. Following sequence verification, the plasmid, along with mCherry-N1 and mEos2-N1 cloning vectors, was sequentially digested with AgeI and NotI and ligated to produce mCherry-Alk5-N-13 and mEos2-Alk5-N-13 fusions.

All DNA for transfection was prepared using the Plasmid Maxi kit (QIAGEN, Valencia, CA) and characterized by transfection in HeLa cells (CCL2 line; ATCC, Manassas, VA) using Effectene (QIAGEN) followed by observation under widefield fluorescence illumination to ensure proper localization. The sequences for all vectors were confirmed using Big Dye technology (The Florida State University Bioanalytical and Molecular Cloning DNA Sequencing Laboratory Tallahassee, FL).

## Cell culture and transfection

Studies were performed using ATDC5 murine chondroprogenitor cells (RCB0565, RIKEN, Wako, Japan), NIH3T3 fibroblasts, MCF10A human mammary epithelial cells, and Human Embryonic Kidney (HEK) 293 cells. Cells were treated as indicated with TGFβ1 (5 ng/ml, HumanZyme, Chicago, IL), Y27632 (10 μM, Sigma-Aldrich, St. Louis, MO), and blebbistatin (10 μM, Cayman Chemical, Ann Arbor, MI).

For imaging experiments, glass-bottom imaging wells (Cellvis, Mountain View, CA ) were coated with collagen II (1 mg/ml in acetic acid diluted 1:100 in PBS), fibronectin (1 mg/ml diluted 1:100 in PBS), or poly-l-lysine (0.1 mg/ml). ATDC5, NIH3T3, and MCF10A cells were transfected using Nucleofection (Lonza, Basel, Switzerland) or Effectene (Qiagen, Valencia, CA), and then plated on to the imaging wells. For biochemical assays, 293 cells were plated in 100 mm cell-culture dishes and transfected using Effectene (Qiagen) at 80% confluency.

## Antibodies, Co-immunoprecipitation, Immunoblotting and Immunofluorescence

For co-immunprecipitation experiments, 293 cells were transfected with integrin αV-mCherry, cofilin-mEmerald, TβRI-Flag, TβRII-Flag, and/or TβRI-Myc. 24 hr after transfection, cells were treated with TGFβ or Y27632 for 15 min. Cells were lysed (50 mM Tris pH 7.5, 150 mM NaCl, 2 mM EDTA, 0.5% Igepal, 0.25% sodium deoxycholate, supplemented with protease and phosphatase inhibitor tablets) and immunoprecipitated with anti-Flag M2 beads (Sigma-Aldrich) overnight prior to Western analysis (*Alliston, 2001*). For immunoblotting experiments of downstream endogenous proteins, 293 cells were cultured on plastic or fibronectin-coated gel substrates (Advanced BioMatrix, Carlsbad, CA), treated with TGFβ as indicated for 30 min and then lysed. Blots were probed with anti-Flag (F3165, Sigma-Aldrich), anti-CD51, integrin αV (611013, BD Biosciences, San Jose, CA), anti-cofilin (ACFL02, Cytoskeleton, Denver, CO), anti-pSmad3 (gift from E. Leof, Mayo Clinic, Rochester, MN), anti-β-actin (ab8226, Abcam, Cambridge, MA) and anti-TβRI (sc-398, Santa Cruz Biotechnology, Santa Cruz, CA), and anti-mouse anti-rabbit secondary antibodies that were conjugated to 680 or 800CW IRDye fluorophores for detection using a LI-COR infrared imaging system (LI-COR Biosciences, Lincoln, NE). Blots shown are representative of multiple technical replicates of at least three independent experiments for each condition (N≥3). Where indicated, quantitative analysis was performed using ImageJ (National Institutes of Health, Bethesda, MD). Band intensity for proteins of interest was normalized to band intensity of controls (Flag for co-IP and β-actin for whole cell extract). Fold change in band intensity was calculated relative to plastic control samples. ANOVA followed by Bonferroni correction and student's *t* test were used to evaluate statistical significance.

For immunofluorescence studies, ATDC5 cells were cultured on collagen II-coated glass substrates in 8 well Lab-Tek chamber slides (Nunc/Thermo Fisher Scientific, Waltham, MA). Cells were fixed (4% paraformaldehyde) and permeabilized (0.5% Triton X-100 in PBS). Primary antibodies included anti-TβRII (sc-1700, sc-400, Santa Cruz) and anti-TβRI (sc-398, sc-9048, Santa Cruz). Cells were imaged as described below.

## Affinity purification and reversed-phase liquid chromatography-electrospray tandem mass spectrometry (LC-MS/MS)

Cells expressing TβRII-Flag or TβRI-Flag and integrin αV-mCherry in 10 cm cell culture dishes were lysed as above and affinity-purified with M2-Flag magnetic beads (Sigma-Aldrich), followed by on-bead trypsin digestion (*Kean et al., 2012*) and mass spectrometry approaches to study associated proteins (N=3). Peptides recovered were analyzed by reversed-phase liquid chromatography-electrospray tandem mass spectrometry (LC-MS/MS) as described (*Duong et al., 2015*). Briefly, peptides were separated by nano-flow chromatography in a C18 column, and the eluate was coupled to a hybrid linear ion trap-Orbitrap mass spectrometer (LTQ-OrbitrapVelos, Thermo Fisher Scientific, Waltham, MA) equipped with a nanoelectrospray ion source. Following LC-MS/MS analysis, peak lists generated from spectra were searched against the human subset of the SwissProt database using in-house ProteinProspector (*Clauser et al., 1999*). For analysis, peptide counts of each protein were normalized by the total protein content in the sample and the molecular weight of the respective protein. This provided an abundance index for each protein that served as a comparison between pulldowns. The ratio between abundance indices for TβR pulldowns to untransfected control (mock) pulldowns was used to screen candidate proteins.

## Image acquisition and analysis

ATDC5, NIH3T3, and MCF10A cells were transiently transfected with fluorescently labeled expression plasmids and plated on collagen II, fibronectin or poly-l-lysine-coated glass-bottom imaging wells. Cells were imaged 24 hr after transfection, and treated with Y27632, blebbistatin, or TGFβ as indicated. Confocal images were obtained on a motorized Yokogawa CSU-X1 spinning disk confocal unit on an inverted microscope system (Ti-E Perfect Focus System, Nikon, Tokyo, Japan), with either a 100X/NA 1.49 oil-immersion objective (CFI Apo TIRF, Nikon) or a 40X/NA 1.15 water-immersion objective (CFI Apo LWD, Nikon), on a front illuminated CMOS camera (Zyla sCMOS, Andor, Belfast, United Kingdom). For TIRF and sptPALM, imaging was performed on a motorized objective-type TIRF inverted microscope system (Ti-E Perfect Focus System, Nikon) with activation and excitation lasers of 405 nm, 488 nm, and 561 nm, and an electron-multiplying charged-coupled device camera (QuantEM 512, Photometrics, Tuscon, AZ), a 100X/NA 1.49 oil-immersion objective (CFI Apo TIRF, Nikon), a stage top incubator (Okolab, Burlingame, CA), and controlled by NIS-Elements software (Nikon). Cells expressing mEos2-tagged constructs were simultaneously activated with a 405 nm laser and excited with a 561 nm laser. Laser intensities were adjusted to maintain a constant sparse population of activated molecules that were spaced well enough for accurate localization and tracking. Prior to each sptPALM imaging sequence and photoconversion of mEos2, the mEmerald signal from mEmerald fusions of vinculin was imaged to localize focal adhesions. NIS-Elements software (Nikon) was used for the acquisition of images at 10 fps. Individual receptors were localized and tracked using a previously described algorithm (*Sbalzarini and Koumoutsakos, 2005*) written in MosaicSuite for ImageJ and available at (www.mosaic.mpi-cbg.de). All images were processed using ImageJ with a 0.6 gaussian blur filter to remove noise. Images shown are representative of multiple cells (N≥5) for at least three independent experiments for each condition.

## Colocalization quantification

TIRF mode imaging was used to obtain intensity profiles of two distinct molecules over adhesion-rich regions of interest (for example, regions shown in *Figure 3A–C*). The similarity of the two profiles was quantified to provide a measure of colocalization, specifically by comparing pixel intensities (8-bit grayscale) at each point across the two profiles. For each pixel, an ordered pair containing the intensities at that particular coordinate from both images was plotted. Values closer to the line y=x refer to coordinates that have very similar intensities in both profiles. Values further from y=x are coordinates that have a mismatch in intensities. By reflecting all points in the top half of this graph across y=x, a distribution of points is created between y=x and the x-axis, but the distance of individual points from y=x is preserved. The magnitude of the slope of the regression line through these points can be used as a quantitative metric of colocalization. The greater this slope, the higher the degree of colocalization. Plots are representative of multiple cells (N≥3) and multiple regions of interest (N=5). ANOVA followed by Bonferroni correction was used to evaluate statistical significance.

## Single-molecule tracking

Each sptPALM imaging sequence generates tens of thousands of molecule trajectories per cell (N=6 cells for each TβR). From these, only trajectories lasting between 0.5 seconds and 2 seconds (5 to 20 frames at 10 fps) were selected for analysis. Tracks that were not confined to either inside or outside focal adhesions were not considered in the quantitative analysis. For each individual track, a series of parameters were calculated to quantify receptor dynamics. These parameters include mean squared displacement (MSD), diffusion coefficient ($D$), and radius of confinement ($r_{conf}$). MSD was computed as per *Equation 1* (*Rossier et al., 2012*):

$$MSD\left(\tau = n \cdot \Delta t\right) = \frac{\sum_{i=1}^{N-n}\left(x_{i+n} - x_i\right)^2 + \left(y_{i+n} - y_i\right)^2}{N - n}. \tag{1}$$

Where $x_i$ and $y_i$ are the coordinates of the molecule at time $i * \Delta t$ and $N$ is the number of frames for which the trajectory persisted. The radius of confinement ($r_{conf}$) of a track is defined to be the magnitude of the radius of the smallest circle that encloses all points in that track. $D$ is defined as one-fourth of the slope of the regression line fitted to the first four values of the MSD as per *Equation 2*.

$$MSD(\tau) = 4D\tau. \tag{2}$$

Using these variables, trajectories were pooled into three fractions: immobile, confined, and freely diffusive. Immobile molecules were defined as being restricted to a radius of confinement equal to one pixel ($r_{conf} < 0.166$ μm). Confined molecules were defined as non-immobile tracks with a diffusion coefficient $D$ of less than 0.2 μm$^2$/s, and the remaining tracks were considered freely diffusive. Custom routines written for Python were used for track quantification, analysis, and visualization (source code). To account for variability in these large data sets consisting of tens of thousands of tracks, we report mean and standard error of the mean (SEM). ANOVA followed by Bonferroni correction and student's *t* test were used to evaluate statistical significance.

To quantify the diffusive behavior of TβRI and TβRII around focal adhesions, we calculated an enrichment ratio to compare track densities in several (N≥5) focal adhesion-rich regions (where vinculin covered more than 25% of total area) across at least three cells. The enrichment ratio was defined as the ratio of the track density (tracks/μm$^2$) inside adhesions to the track density outside adhesions within a given area. Student's *t* test was used to evaluate statistical significance.

## Statistical analysis

For colocalization quantification (*Figure 3* and *Figure 6*) and enrichment ratio (*Figure 4C*), we report mean and standard deviation (SD). There are three circumstances in which it was more statistically appropriate to report the standard error of the mean (SEM). Specifically, to account for variability in large sptPALM data sets (consisting of tens of thousands of tracks), we report mean and SEM (*Figure 2E,F* and *Figure 4I*). Significance was calculated with ANOVA followed by Bonferroni correction and student's *t* test, with significance defined as p<0.01.

## Acknowledgements

The authors wish to thank V Weaver, J Lakins, and the Weaver lab members for their important intellectual discussions and contributions, K Thorn and D Larsen from the Nikon Imaging Center and T Wittmann for sharing their expertise and assisting with microscopy studies, and E Leof for the phospho-Smad3 antibody. Mass spectrometry analysis was provided by the Bio-Organic Biomedical Mass Spectrometry Resource at UCSF (AL Burlingame, Director) supported by funding from the Biomedical Technology Research Centers program of the NIH National Institute of General Medical Sciences, NIH NIGMS 8P41GM103481, and Howard Hughes Medical Institute. This work was supported by National Science Foundation Graduate Research Fellow Program Grant No. 1144247 (JR), Department of Defense through the National Defense Science and Engineering Graduate Fellowship Program (DM), National Institute of Arthritis, Musculoskeletal and Skin Disease R21 AR067439-01 (TA), and National Institute of Dental and Craniofacial Research R01 DE019284 (TA). Any opinions, findings, and conclusions or recommendations expressed in this material are those of the author and do not necessarily reflect the views of the National Science Foundation.

# Additional information

## Funding

| Funder | Grant reference number | Author |
|---|---|---|
| National Science Foundation | Graduate Research Fellowship, 1144247 | Joanna P Rys |
| American Society for Engineering Education | National Defense Science and Engineering Graduate Fellowship Program | David A Monteiro |
| National Institute of General Medical Sciences | 8P41GM103481 | Alma L Burlingame |
| Howard Hughes Medical Institute | | Alma L Burlingame |
| National Institute of Arthritis and Musculoskeletal and Skin Diseases | R21 AR067439-01 | Tamara N Alliston |
| National Institute of Dental and Craniofacial Research | R01 DE019284 | Tamara N Alliston |
| U.S. Department of Defense | OR130191 | Tamara N Alliston |

The funders had no role in study design, data collection and interpretation, or the decision to submit the work for publication.

## Author contributions

JPR, CCDuF, DAM, Conception and design, Acquisition of data, Analysis and interpretation of data, Drafting or revising the article; MAB, MWD, Drafting or revising the article, Contributed unpublished essential data or reagents; JAOP, Acquisition of data, Analysis and interpretation of data, Drafting or revising the article; SC, Acquisition of data, Analysis and interpretation of data; ALB, Conception and design, Analysis and interpretation of data; TNA, Conception and design, Analysis and interpretation of data, Drafting or revising the article

## Author ORCIDs

Juan A Oses-Prieto, http://orcid.org/0000-0003-4759-2341

# Additional files

## Supplementary files

• Source code 1. sptPALM scripts. Python cripts to help with data analysis, specifically for sptPALM analysis and visualization (*Figures 2* and *4*)

• Source code 2. Colocalization quantification. *Figure 3Ai-Ci* and *Figure 6Ai-Di.*

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
