## [Decision Letter]

Thank you for submitting your work entitled "Discrete spatial organization of TGFβ receptors couples receptor multimerization to cellular tension" for peer review at *eLife*. Your submission has been favorably evaluated by Fiona Watt (Senior editor) and three reviewers, one of whom is a member of our Board of Reviewing Editors.

The reviewers have discussed the reviews with one another and the Reviewing editor has drafted this decision to help you prepare a revised submission.

All of the three referees found the manuscript potentially interesting. However, Referee #1 and #2 raised very similar concerns about the lack of data showing that the segregation of the receptors actually regulates TGFβ signalling in cells. They were also concerned about the validity of the data with respect to endogenous receptors as all the data provided is with overexpressed receptors. An additional point of concern was that the mechanosensitivity has only been addressed with cytoskeleton modifying drugs and not for example using engineered matrixes of different stiffness. Finally, quantification and statistic needs to be improved.

Essential revisions:

1) To warrant publication in a high-quality journal like *eLife* the authors should expand their experiments on cellular tension and the plausible mechanosensitivity of the TGFβ receptor heterodimerisation beyond the use of ROCK inhibitor and blebbistatin. For example some of the following approaches should be tested:

What happens to the organization of the receptors when cells are spread on different size micropatterns with equal adhesive area?

What is the contribution of integrin activity to the phenomena – this could be tested by comparing cells plated on surfaces coated with active β1-integrin antibody like 12G10 vs. inactive integrin antibody like 4B4. Is the receptor segregation lost when cells are plated on integrin-independent surface like poly-L-lysine?

TIRF imaging and sptPALM are not compatible with cells plated on supports (hydrogels, PAGE-gels) of variable stiffness. However, the co-IP experiments and possible TGFβ signalling experiments could be extended to cells growing on soft and stiff supports. Furthermore, confocal microscopy could be attempted.

2) The experiments need to be better explained (e.g. time courses of the experiments performed, concentrations of matrices used, rationale for selecting only and exclusively αV and α2 containing integrins).

3) Thus far all the data provided are with over-expressed proteins. It is important to validate the TIRF data with staining of endogenous proteins to demonstrate that the sequestration is not an artifact of tagged proteins or overexpression.

4) Some data are over interpreted. The authors talk about physical association between TβRI and integrin αV or TβRI and integrin αV. Given that the IPs presented were performed on cell lysates of overexpressing cells, then the authors can only conclude that TβRI and TβRII form a complex with integrins, but whether this is a direct contact needs to be addressed by better protein-protein interactions assays.

5) Importantly, it is not clear what the significance of the study proposed truly is. The authors show that cellular tension might regulate multimerization of TβRI and TβRII, but whether this has any implication in receptor activation, downstream signaling and, importantly, cell function is highly debatable. The authors finish the paper suggesting that "…which would be required to activate downstream signalling". This should be tested experimentally.

[Editors' note: further revisions were requested prior to acceptance, as described below.]

Thank you for resubmitting your work entitled "Discrete spatial organization of TGFβ receptors couples receptor multimerization and signaling to cellular tension" for further consideration at *eLife*. Your revised article has been favorably evaluated by Fiona Watt (Senior editor) and a Reviewing editor. The manuscript has been improved but there are some remaining issues that need to be addressed before acceptance, as outlined below:

The data regarding the sensitivity of endogenous TGFβ to stiffness of the matrix are interesting and important for the study. While they address most of the points raised by the reviewers it seems that in their current form they fall short of the scientific quality expected for *eLife*. The figure legends do not indicate how many times these experiments have been performed. The gold standard would be 3 and for statistics 4 would be recommended. Please repeat these experiments if necessary and provide quantification from 3-4 independent biological replicates to validate the claims for increased co-precipitation and altered SMAD-signalling.

---

## [Author Response]

All of the three referees found the manuscript potentially interesting. However, Referee #1 and #2 raised very similar concerns about the lack of data showing that the segregation of the receptors actually regulates TGFβ signalling in cells. They were also concerned about the validity of the data with respect to endogenous receptors as all the data provided is with overexpressed receptors. An additional point of concern was that the mechanosensitivity has only been addressed with cytoskeleton modifying drugs and not for example using engineered matrixes of different stiffness. Finally, quantification and statistic needs to be improved.

Essential revisions:

1) To warrant publication in a high-quality journal like eLife the authors should expand their experiments on cellular tension and the plausible mechanosensitivity of the TGFβ receptor heterodimerisation beyond the use of ROCK inhibitor and blebbistatin. For example some of the following approaches should be tested:

What happens to the organization of the receptors when cells are spread on different size micropatterns with equal adhesive area?

What is the contribution of integrin activity to the phenomena – this could be tested by comparing cells plated on surfaces coated with active β1-integrin antibody like 12G10 vs. inactive integrin antibody like 4B4. Is the receptor segregation lost when cells are plated on integrin-independent surface like poly-L-lysine?

TIRF imaging and sptPALM are not compatible with cells plated on supports (hydrogels, PAGE-gels) of variable stiffness. However, the co-IP experiments and possible TGFβ signalling experiments could be extended to cells growing on soft and stiff supports. Furthermore, confocal microscopy could be attempted.

We agree with the importance of examining the mechanoregulation of TGFβ receptor heteromerization using multiple approaches. We have performed several experiments to address this point, including those helpfully suggested by the reviewers. A variety of imaging and biochemical outcomes collectively yield new data that strongly support the regulation of TGFβ receptor heteromerization and downstream signaling by cellular tension (Figure 6 and Figure 7).

To determine the effect of integrin adhesion on TβR localization, we examined integrin α2-mCherry and TβRII-mEmerald localization in ATDC5 cells grown on substrates coated with collagen II, poly-l-lysine, or integrin blocking antibodies. Cells were imaged with spinning disc confocal and TIRF microscopy (Figure 3—figure supplement 1). TβRII was excluded from integrin-rich focal adhesions in cells grown on collagen II-coated substrates. On the other hand, when grown on poly-l-lysine-coated substrates where binding does not require integrins, cells do not form adhesions and show no evidence of TβRII exclusion. We observed identical but less striking results when cells were plated on substrates coated with integrin inhibitory (4B4) antibodies (not shown). These findings indicate that integrin-mediated adhesion is required for the unique spatial organization of TβRII, and that this organization is maintained with high cellular tension.

The strongest support for mechanoregulation of TβR heteromerization has come from new biochemical outcomes that correspond to the increased TβRII/integrin α2 colocalization observed following ROCK or myosin II inhibition (Figure 6). First, with the reduction in cellular tension following ROCK inhibition, TβRI preferentially co-precipitates with its signaling partner TβRII (Figure 7). Second, a reduction in cellular tension by culture of cells on compliant substrates also promotes co-precipitation of TβRI with TβRII (Figure 7). Third, a reduction in cellular tension by culture of cells on compliant substrates drives Smad3 phosphorylation (Figure 7, Allen et al., 2012). Finally, the tension-dependent changes in TβRI/TβRII multimerization and Smad3 phosphorylation correspond to changes in TGFβ-inducible Smad nuclear translocation (not shown, Allen et al., 2012). Therefore, these data extend and further strengthen the conclusion that cellular tension controls the spatial organization and formation of active TβRI/TβRII complexes as well as the phosphorylation and nuclear localization of the key TGFβ effector Smad3.

Several other approaches to examine the effect of cellular tension on TβR localization did not yield definitive results, largely due to technical limitations. As the reviewers indicate, TIRF imaging on gels is extremely difficult because the refractive index is not matched to that of glass. Confocal imaging could not definitively discriminate the effect of cellular tension on TβR localization in cells grown on variable stiffness substrates (polyacrylamide or PDMS). We also manipulated cellular tension by varying the size, shape, and surface area of micropatterned adhesive substrates acquired from collaborators and from commercial sources. Although imaging revealed cellular adhesions on each of these substrates, the resolution was insufficient to determine the effect of cellular tension on TβR spatial organization. Significant additional technical and scientific effort, beyond the scope of the current project, is needed to successfully use TIRF to determine the effect of these micropatterns on TGFβ receptor localization.

2) The experiments need to be better explained (e.g. time courses of the experiments performed, concentrations of matrices used, rationale for selecting only and exclusively αV and α2 containing integrins).

We revised the Methods to include the requested experimental details. We have revised the Results to explain the rationale for evaluating integrins α2 and αV. Integrins α2 and αV were chosen since they are primary integrins in chondrocytes, where they bind collagen and vitronectin/fibronectin, respectively. Furthermore, both of these integrins have been shown to interact with the TGFβ pathway. For example, integrin αV activates TGFβ ligand from its latent form through generation of cellular tension (Wipff et al., 2007), and integrin α2β1 physically associates with TGFβ receptors during collagen-induced Smad phosphorylation (Garamszegi et al., 2010; Introduction). It will be interesting to further expand these studies to examine other integrin members as well as other cell types in the future.

3) Thus far all the data provided are with over-expressed proteins. It is important to validate the TIRF data with staining of endogenous proteins to demonstrate that the sequestration is not an artifact of tagged proteins or overexpression.

We share the reviewers’ concern and have worked hard to demonstrate the relevance of these findings to the endogenous TGFβ signaling pathway. The revised manuscript addresses this concern with new data, including mass spectrometry for endogenous TβR interacting proteins and assays for activation of endogenous TGFβ effectors (Figure 5 and Figure 7).

As suggested, we tried to augment the manuscript with data showing the spatial localization of endogenous TGFβ receptors. However, we were limited by the well-known lack of suitable reagents for immunofluorescent detection of endogenous TβRI or TβRII. Indeed, few published studies show TβR IF. Those that do are unable to achieve the resolution apparent in studies by our group and others that use fluorescently-tagged TβRs (Ma et al., 2007). Our attempts to visualize endogenous TβRs with two different TβRII antibodies and two different TβRI antibodies showed specificity relative to IgG controls and were comparable to published results (Figure 1 and Figure 1—figure supplement 1; Garamszegi et al., 2010). Unfortunately, this bright punctate staining was inadequate to observe finer spatial organization. Furthermore, this approach is limited by the inability to perform dynamic imaging, which is essential for the fundamental understanding of these receptor-receptor interactions and their responses to physical or biological stimuli (Video 1, Video 2, and Video 3). In spite of the inability to visualize the spatial organization of endogenous TβRs, the implication of endogenous Smad3 in tension-dependent regulation of TGFβ signaling (Figure 7) gives confidence that these findings are physiologically relevant.

Figure 5 shows the results of mass spectrometry analysis of TβRI/TβRII interacting proteins that may participate in their spatial organization at focal adhesions. In addition to anticipated identification of co-expressed integrin αV, TβRII binds to endogenous cofilin, an actin-binding protein involved in the regulation of actin filaments and in cell motility. The peptide counts indicate that endogenous cofilin preferentially associates with TβRII relative to TβRI, a finding supported by co-IP assays (Figure 5). Future studies will examine additional candidates that may contribute to this discrete spatial organization and explore the functional role of cofilin in the mechanoregulation of TβR heteromerization and function.

Figure 7 shows the tension-sensitive regulation of endogenous Smad3 phosphorylation by cellular tension and TGFβ. We performed the same experiment in cells with and without overexpressed TβRI/TβRII and achieved the same results (not shown). This evidence that cellular tension regulates the endogenous TGFβ signaling pathway complements our prior observations of stiffness-dependent regulation of endogenous TGFβ1 and chondrocyte marker gene expression, as well as the phosphorylation and nuclear localization of endogenous Smad3 (Allen et al., 2012). These findings demonstrate the capacity for cellular tension to modulate the activity of the endogenous TGFβ signaling pathway.

4) Some data are over interpreted. The authors talk about physical association between TβRI and integrin αV or TβRI and integrin αV. Given that the IPs presented were performed on cell lysates of overexpressing cells, then the authors can only conclude that TβRI and TβRII form a complex with integrins, but whether this is a direct contact needs to be addressed by better protein-protein interactions assays.

We agree that additional studies are required to conclude that the interaction of TβRs with integrins is direct. We revised the text to clearly communicate that this interaction may be indirect (Results). Future experiments will involve assays such as FRET and BRET to confirm the interactions between TβRs and integrins. Receptors tagged with validated FRET pairs, such as mTurquoise and mNeonGreen (Shaner et al., 2013), will be used to monitor the dynamics and proximity of protein interactions inside living cells.

5) Importantly, it is not clear what the significance of the study proposed truly is. The authors show that cellular tension might regulate multimerization of TβRI and TβRII, but whether this has any implication in receptor activation, downstream signaling and, importantly, cell function is highly debatable. The authors finish the paper suggesting that "…which would be required to activate downstream signalling". This should be tested experimentally.

We found that manipulation of cellular tension through ROCK inhibition or culture on variable stiffness substrates regulates TGFβ receptor multimerization. Specifically, culturing cells on plastic with Y27632 or culturing cells on a soft 0.5 kPa substrate drives TβRI/TβRII complex formation. These conditions correspond to TGFβ-inducible Smad3 phosphorylation. This suggests that cellular tension can structurally prime TGFβ receptors to respond to the addition of TGFβ, consistent with previous findings (Allen et al., 2012). Overall, these new data strengthen our finding of a novel mechanism by which cellular tension regulates the spatial organization and downstream signaling of TGFβ.

[Editors' note: further revisions were requested prior to acceptance, as described below.]

The data regarding the sensitivity of endogenous TGFβ to stiffness of the matrix are interesting and important for the study. While they address most of the points raised by the reviewers it seems that in their current form they fall short of the scientific quality expected for eLife. The figure legends do not indicate how many times these experiments have been performed. The gold standard would be 3 and for statistics 4 would be recommended. Please repeat these experiments if necessary and provide quantification from 3-4 independent biological replicates to validate the claims for increased co-precipitation and altered SMAD-signalling.

1) We carefully reviewed all of our Western/IP analyses, especially those evaluating the effect of substrate stiffness on TβRI/TβRII complex formation and Smad phosphorylation. Indeed, each experiment was repeated independently a minimum of 3 times. The Methods, rather than the figure legends, were updated where needed to consistently describe and define in more detail the number and type of replicates used for all studies, as follows:

“Blots shown are representative of multiple technical replicates of at least three independent experiments for each condition (N≥3).”

2) Because of the importance of the conclusions in Figure 7 regarding the effect of substrate stiffness on TGFβ receptor multimerization and Smad3 phosphorylation, we appreciate the opportunity to rigorously document these findings. Prior to our first resubmission, we had performed these experiments three independent times, but did not quantify the results. We added another biological replicate for experiments performed on substrates of varying stiffness, for a total N≥4. We see that the increase in TGFβ receptor multimerization on the 0.5 kPa substrate is qualitatively reproducible and quantitatively increased across all 5 biological replicates, with a significant difference of p < 0.05 (N=5).

Similarly, the stiffness-dependent regulation of basal and TGFβ-inducible Smad3 phosphorylation is reproducible and visually evident across multiple biological and technical replicates. In addition, the increase in Smad3 phosphorylation on the 0.5 kPa substrate, without and with added TGFβ, is statistically significant relative to the plastic control (p < 0.05, N=4). In some cases, the apparent qualitative differences are reflected in the quantitative analyses, even though they do not achieve statistically significance. We believe that the conclusions described in the Results are supported by the Figures and by the addition of a new Source Data file showing the raw data used for the quantitative analyses in Figure 7. The details of this quantitative analysis have been added to the Methods, as follows:

“Where indicated, quantitative analysis was performed using ImageJ. Band intensity for proteins of interest was normalized to band intensity of controls (Flag for co-IP and β-actin for whole cell extract). Fold change in band intensity was calculated relative to plastic control samples. ANOVA followed by Bonferroni correction and student’s t-test were used to evaluate statistical significance.”

In addition, the p value has been added to the legend for Figure 7, and the word ‘significantly’ has been added to the corresponding section of the Results, as follows:

Figure 7 Legend: “Likewise, manipulation of cellular tension through culturing cells on collagen II-coated glass or 0.5 kPa PDMS substrates increases co-immunoprecipitation of TβRI with Flag-tagged TβRII (p < 0.05, B). In cells grown on collagen II-coated compliant (0.5 kPa, p < 0.05) or stiff (16 kPa) PDMS substrates, endogenous Smad3 phosphorylation is increased (C). The effect of TGFβ on Smad3 phosphorylation is substrate-dependent, such that maximal TGFβ-inducibility is observed on 0.5 kPa substrates (p < 0.05), consistent with a tension-sensitive calibration of TβR localization and activity (C). See [Supplementary-material SD5-data].”

Results: “A reduction in cellular tension through culture on compliant substrates *significantly* drives TβRI/TβRII complex formation (Figure 7). Culturing cells on compliant substrates leads to *significantly* increased endogenous Smad3 phosphorylation (Figure 7).”